# Evaluation of the Effect of Ultrasonic Pretreatment on the Drying Kinetics and Quality Characteristics of *Codonopsis pilosula* Slices Based on the Grey Correlation Method

**DOI:** 10.3390/molecules28145596

**Published:** 2023-07-24

**Authors:** Tongxun Wang, Xinyu Ying, Qian Zhang, Yanrui Xu, Chunhui Jiang, Jianwei Shang, Zepeng Zang, Fangxin Wan, Xiaopeng Huang

**Affiliations:** College of Mechanical and Electrical Engineering, Gansu Agricultural University, Lanzhou 730070, China; wangtx@st.gsau.edu.cn (T.W.); yingxy@st.gsau.edu.cn (X.Y.); zhangq@st.gsau.edu.cn (Q.Z.); yangh@st.gsau.edu.cn (Y.X.); jiangch@st.gsau.edu.cn (C.J.); shangjw@st.gsau.edu.cn (J.S.); zangzp@st.gsau.edu.cn (Z.Z.); wanfx@gsau.edu.cn (F.W.)

**Keywords:** pretreatment, *Codonopsis pilosula*, drying characteristics, quality, grey correlation

## Abstract

Ultrasonic (US) maltreatment was performed before the vacuum far-infrared drying (VFID) of *Codonopsis pilosula* (CP) slices to investigate the effects of different US parameters on the drying characteristics and nutrients of CP slices. The grey correlation method with relative correlation degree (*r_i_*) as the evaluation measure was used to construct a model for the evaluation of the pretreatment quality of CP and to determine the optimal pretreatment conditions. The results showed that with the increase in US frequency and power, the drying rate increased. Under the conditions of US power of 180 W, frequency of 60 kHz and a pre-treatment time of 30 min, the drying time reduced by 28.6%. The contents of polysaccharide and syringin in dried CP slices pretreated by US increased by 14.7% and 62.0%, respectively, compared to the non-pre-treated samples, while the total flavonoid content decreased by 10.0%. In terms of colour, pretreatment had a certain protective effect on the red colour of dried products. The highest relative correlation (0.574) and the best overall quality of performance were observed at 180 W, 60 kHz and 30 min. Overall, US technology is suitable for the pretreatment processing of CP, which is of great significance to the drying of CP.

## 1. Introduction

*Codonopsis pilosula* (CP) is a perennial herb; it is the dried root of CP (Franch.) Nannf, *C. tangshen* Oliv. and *C. pilosula* Nannf. var. modesta (Nannf.) [1]. CP is an important functional food containing polysaccharides, triterpenoids, sesquiterpenes, phenolic glycosides, alkaloids and other bioactive components [2,3,4]. It has the effect of enhancing immunity, relieving physical fatigue, assisting in protecting gastric mucosa, causing antioxidation and so on [5], it is widely used in nutraceuticals [6], and it has a high nutritional value.

As the moisture content of fresh CP is difficult to control, and the initial processing and storage are prone to moisture absorption, mildew, oil, moth and other phenomena, it is difficult to effectively preserve the active ingredient [7,8] and it needs to be dried. Since the drying process involves the heat and mass transfer of materials, the content of active ingredients is greatly affected by the drying process. Therefore, a reasonable drying method is needed to reduce the loss of active ingredients after drying. At present, the commonly used modern drying methods of CP include microwave, hot air, infrared, ultrasonic-assisted methods [9], etc., and although various drying methods have advantages and disadvantages, they generally fail to meet the ideal industrial processing requirements [10].

Vacuum far-infrared drying (VFID) combines the advantages of vacuum low-temperature anaerobic and far-infrared radiation non-contact heating to effectively retain the heat-sensitive active ingredients in the material, and can significantly increase the drying rate and shorten the drying time [11]. Vacuum far-infrared technology has been applied to the drying of banana slices [12], potato slices [13], aquilaria leaves [14] and other related research, showing advantages of low energy consumption, high drying rate and good quality dried products. Pretreatment before drying can effectively strengthen the heat and mass transfer process of the material, improve its drying efficiency, and reduce the quality deterioration of the final product [15]. As pre-treatment technology for drying, ultrasonic (US) cavitation and sponge effects [16,17] involve forces that form microscopic channels in the material tissue, reducing the resistance to moisture migration inside the material and promoting moisture diffusion [18]. Related studies have shown that the US pretreatment of saffron for far-infrared drying can not only shorten the drying time, but also retain the colour and aroma characteristics, antioxidant activity and flavonoid content efficiently [19]; the ultrasonic pretreatment of ginkgo seeds prior to infrared drying had a positive effect on the drying process, with a significant increase in the total phenolic content and antioxidant activity, and the lowest stored concentration of toxic components [20].

The purpose of this study was to evaluate the effect of US pretreatment on the drying kinetics and quality characteristics of CP, including colour difference, microstructure, and functional composition. A quality evaluation model for US CP pretreatment was constructed using grey correlation analysis to obtain the optimum drying conditions using the relative correlation degree as a measure. Pretreatment technology before the drying of CP is of great importance to improve the processing efficiency and product quality.

## 2. Results

### 2.1. Analysis of Drying Characteristics

#### 2.1.1. Effect of US Frequency on Drying Characteristics

The effect of US frequency on the VFID characteristics of CP slices is shown in Figure 1. It can be seen that the control group took the longest time to dry (280 min), indicating that US pretreatment accelerated the drying process and shortened the drying time. The drying time of the CP slices was reduced by 25.0%, 28.6% and 32.1% at US frequencies of 20, 40 and 60 kHz, respectively, compared to the control group, because the ultrasonic action expanded the pores of the internal tissues of CP slices, increased the microchannels, and reduced the transmission resistance of moisture loss [21]. The drying rate curve indicated that the drying rate of CP slices increased with the increase in US frequency. This may be due to the fact that the higher the US frequency, the smaller its vibrational period, and the bubbles produced by its cavitation break up, generating high temperature and pressure, promoting the rate of heat mass transfer and accelerating the movement of water within the CP.

#### 2.1.2. Effect of US Power on Drying Characteristics

The effect of US power on the VFID characteristics of CP slices is shown in Figure 2. The time required to reduce the moisture content of the dry base of the materials to 0.10 ± 0.005 g/g at different US powers was 210 min, 200 min and 190 min, respectively, while the control group had the longest drying time. It can be seen that the effect of US power on the drying time was more significant. Observation of the drying rate curves revealed a brief rise in drying rate at the beginning of the drying period, which then shifted to a distinct descending section, with a slow decline at the end of the drying period, and similar results were obtained in studies of the US pretreatment of sliced onion stems [22] and dried sweet potatoes [23]. The drying rate of CP slices increased with the increase in US power, and this result may be related to the change in moisture state and distribution of CP slices. With the increase in US power, the free degree of free water and bound water inside CP increased, the mobility was enhanced, the binding of solids and ions inside CP decreased, and the drying rate increased [24].

#### 2.1.3. Effect of Pre-Treatment Time on Drying Characteristics

The effect of US pre-treatment time on the VFID characteristics of CP slices is shown in Figure 3. The longer the pre-treatment time, the shorter the time required to dry the CP slices to below the safe moisture content. This may be due to the increased effect of cavitation and sponge effect caused by the prolongation of the US pre-treatment time on the internal moisture state and distribution of CP slices, thus accelerating the migration and removal of moisture during the VFID of CP slices. The analysis of the drying rate curves showed that the drying rate of CP increased with the increase in US pre-treatment time, which may be related to the enhancement of mechanical and cavitation effects generated by ultrasound. In addition, the US effect produced more microporous channels inside the material to increase the effective water diffusion coefficient of the water molecules, which reduced the mass transfer resistance and promoted the flow of water from the interior to the surface of CP [25], thus achieving enhanced water diffusion in the VFID process of CP slices; similar results were obtained from Alizehi’s study on carrots [26].

### 2.2. Effect on Effective Moisture Diffusivity

The effective moisture diffusivity (*D_eff_*) of VFID of CP slices under different US pretreatment conditions is shown in Table 1, with *D_eff_* ranging from 2.399 × 10^−8^ to 2.775 × 10^−8^ m^2^/min, which is consistent with the processing range of agricultural products (10^−8^ to 10^−12^), and is consistent with the results of Zogzas’ study [27]. Compared to the control group (1.764 × 10^−8^ m^2^/min), the *D_eff_* of the material increased by 43.4% after US pretreatment, which may be due to the mechanical effect of the ultrasound, which ruptured the cavitation bubbles and caused a series of physical and chemical effects resulting in an increase in the *D_eff_* of the material [28]. In addition, the *D_eff_* of CP slices increased correspondingly with the increase in US frequency and power. This is because the enhanced ultrasound action induced an increase in the number of pores in the microcapillaries, accompanying moisture movement and mass transfer, which accelerated the rate of moisture diffusion [29] and accelerated the drying process.

### 2.3. Study on the Drying Quality of CP

#### 2.3.1. Effect on Total Flavonoid Content

The effect of US pretreatment under different parameters on the total flavonoid content of dried CP slices is shown in Figure 4a; the figure shows that US pretreatment will reduce the total flavonoid content in CP slices, with an average content of 1.036 mg/g, which is 10.0% lower than that of the control group (1.151 mg/g). This may be due to the fact that US processing promotes the extraction procedure of the target substances, leading to the release of flavonoids, which are susceptible to oxidation due to their abundant hydroxyl groups [19]. The variance analysis showed that the total flavonoid content was significantly different under different US powers and pre-treatment times (*p* < 0.05). The total flavonoid content tended to increase and then decrease with increasing sonication time, reaching the highest value (1.062 mg/g) at 30 min, indicating that longer sonication was not conducive to the retention of total flavonoids. Total flavonoid content showed a decreasing trend with increasing US power, which may be due to the fact that high intensity ultrasound is more likely to break the covalent bonds between polymers and promote the release of flavonoids; however, the released flavonoids are easily oxidized and cause loss [30]. Therefore, the optimum pretreatment conditions for obtaining total flavonoids are 180 W US power, 60 kHz US frequency and 30 min pre-treatment time.

#### 2.3.2. Effect on Polysaccharide Content

The effect of different pretreatment conditions on the polysaccharide content of CP is shown in Figure 4b. Compared to the control sample (39.559 mg/g), the mean value of polysaccharide content in CP slices (45.354 mg/g) increased by 14.7% after US pretreatment. When the US frequency was 60 kHz, the polysaccharide content was highest (50.528 mg/g). The ANOVA results showed significant differences between the pretreatment powers (*p* < 0.05), indicating that the different US pretreatment power conditions had a greater effect on the polysaccharide content of the dried products. The polysaccharide content increased with the increase in US power and then decreased, reaching the highest level (46.425 mg/g) at 180 W. This may be due to the high power accelerating the rupture of cavitation vesicles, with small cracks in the cells and larger cell gaps reducing the amount of polysaccharides in the sample. In addition, the polysaccharide content of CP increased with the increase in ultrasonic treatment time. This may be due to the increase in US pre-treatment time, which increased the drying rate of CP, shortened the drying time and reduced the loss of polysaccharides during VFID process [31].

#### 2.3.3. Effect on Total Phenolic Content

The effect of different US pretreatment conditions on the total phenolic content is shown in Figure 4c, where the total phenolic content of the VFID dried samples increased significantly after US pretreatment compared to the fresh samples (0.529 mg/g). However, the difference between the US-pretreated and control samples was not significant, and a similar variation was observed in the study of persimmon samples [32]. From the test results of the total phenolic content, it was clear that the higher the ultrasound intensity, the stronger the thermal effect, accelerating the thermal movement of molecules such as phenols, resulting in a decrease in the total phenolic content with an increase in US power. The total phenolic content showed a trend of increasing and then decreasing with the extension of the pre-treatment time, reaching the highest level (0.826 mg/g) at 30 min of action, but lower than the total phenolic content in the slices of the control group without pretreatment, indicating that the retention of total phenolic substances in the slices of CP could be increased by appropriately prolonging the pre-treatment time. It is possible that the ultrasound has a better effect on the cellular integrity and inhibits the release of phenolic components [23].

#### 2.3.4. Effect on Antioxidant Capacity

As DPPH radicals have single electrons, their alcohol solutions are purple in colour. When antioxidants are present, the single electrons of DPPH are scavenged and the colour of the solution becomes lighter, with a linear relationship between its discolouration and the degree of scavenging. Therefore, the antioxidant capacity of CP was evaluated according to inhibition rate. The greater the inhibition rate, the stronger the antioxidant capacity. From Figure 4d, it can be seen that at 150 W of US power, the inhibition of the dried sample was 48.68%, an increase of 6.7% over the control sample. This indicates that the US pretreatment was able to improve the antioxidant capacity of the material, which may be a result of the chemical reaction promoted by US. In addition, the hydroxyl groups produced by ultrasonic cavitation can also contribute to the antioxidant capacity of the material [33], such as the hydroxyl derivatives produced by flavonoids, which can increase the antioxidant activity of the material [34]. This is consistent with the trend in the total flavonoid content of the pretreated dried product extracts, which may be the main reason for the higher antioxidant activity. Other studies have shown that CP polysaccharide is one of the main components with antioxidant activity in CP extract, which improves the antioxidant activity [35]. With the enhancement of US power, the antioxidant ability of the material is weakened, which may be because, with the increase in power, the cavitation effect of ultrasound is enhanced, producing a more active nature and strong oxidizing radicals, and the stability of the material is reduced, resulting in a reduced resistance to oxidation [36].

#### 2.3.5. Analysis of Lobetyolin, Syringin and Atractylenolide III Using HPLC

A sample of 10 μL was injected according to the chromatography conditions and linear fit was performed on the mass concentrations and peak areas of the mixed controls. The mass concentration was the horizontal coordinate (X, mg) and the peak area (Y) was the vertical coordinate. The results are shown in Table 2.

Figure 5 shows the changes in the content of functional substances in dried CP under different US pretreatment parameters; the amounts of lobetyolin, atractylenolide III and syringin in different dried samples were significantly different.

In the VFID process, US pretreatment has a greater impact on the content of syringin. Compared with the control group (0.032 mg·g^−1^), the average syringin content (0.053 mg·g^−1^) after US pretreatment increased by 62.0%. When the ultrasonic power increased from 180 W to 210 W, the content of syringin increased slightly. The content of lobetyolin decreased with the increase in US pretreatment time, which indicates that too long a pretreatment time will lead to the loss and degradation of lobetyolin in the drying process [37]. Under different US power, the content of lobetyolin reached the maximum value of 2.009 mg·g^−1^ at a US power of 210 W. This showed that the retention of lobetyolin was higher at higher ultrasonic power. Ultrasonic power had a slight effect on the content of atractylenolide III. The content of atractylenolide III in the dried samples after US pretreatment was higher than that in the control group, and the content decreased to 0.246 mg·g^−1^ when the pretreatment time increased to 40 min. Upon comprehensive analysis, the optimal conditions of pretreatment were US power 180 W, ultrasonic frequency 40 kHz and pre-treatment time 20 min; under these condition, there was a positive effect on the retention of the active ingredients of CP.

### 2.4. Colour Evaluation

The effects of different US pretreatment conditions on the colour change of CP slices are shown in Table 3. It can be seen that the total colour difference ∆*E* value of all pretreated samples was lower than that of the control sample and closer to the fresh sample. The ∆*E* of CP slices was the lowest (5.64) when the US power was 180 W, the ultrasonic frequency was 40 kHz and the pre-treatment was 40 min; this may be because the increase in pre-treatment time can effectively prevent the excessive oxidation of CP in the drying process and increase the brightness. As the ultrasound frequency increased, the degree of damage to the cell wall became more dramatic, thus accelerating browning and leading to an increase in ∆*E*. The brightness *L** values of the US pretreated samples were on the white side, which may be due to the shrinkage and structural deformation during the drying process, which would transfer photons or absorb more brightness [38]. The *a** value of US pretreatment samples after drying was mostly higher than that of non-pretreatment samples (0.48). This shows that the application of US pretreatment before the drying process had a certain protective effect on the red colour of the product.

### 2.5. Microstructural Analysis

The scanning electron microscope images of CP dried with VFID after US pretreatment are shown in Figure 6. It can be seen that there were obvious differences in the microstructure of CP slices under different parameters. The microstructures of natural drying and control samples had characteristics of small holes and dense structures. The integral cell structure was obvious in the dried CP slices pretreated by US pretreatment, and micropores were formed on the surface; similar structural changes were also observed in pineapple [39] and strawberry slices [40]. The formation of micropores in the sample may be due to the cavitation and sponge effect of the ultrasound, resulting in the loss of cell adhesion and the formation of large intercellular space [41]. The number of micropores increased with the prolongation of the pretreatment time and the increase in ultrasonic power, but high frequency would cause cracks and gaps in the cells. When the US frequency was 20 kHz, the pores of the sample were small and the intercellular space was irregular. When the US frequency was 60 kHz, cracks appeared in the cells and the cell gap became larger. The longer the pre-treatment time, the better the cellular integrity and the uniform size of the micropore channels.

### 2.6. Quality Evaluation of CP

The grey model dataset for evaluating the quality of CP was established using the drying time, *D_eff_* value, polysaccharides, total flavonoids, total phenols, antioxidant activity, colour difference, syringin, lobetyolin and atractylenolide III content of CP determined at different levels of test factors. The results are shown in Table 4, and the 10 evaluation indexes of each group of samples are one evaluation unit.

According to the definition of relative correlation degree, the relative correlation degree of each sample after drying was calculated and sorted according to its size, as shown in Table 5. The relative correlation degree ri of the evaluation model established by 10 indicators was 0.383~0.574, according to the *r_i_* of the dried samples, and the combined effect of different pretreatment conditions on the drying quality was ranked as follows: 210 W < 20 kHz < 40 min < 30 min < 20 min < 150 W < 60 kHz. The ultrasonic frequency of 60 kHz was 0.685 relative to the optimal reference sequence *r_i(s)_* and 0.509 relative to the worst reference sequence *r_i(t)_*, and the comprehensive quality was better. There were some differences in their relative correlation, indicating that the quality of CP was different under different pretreatment conditions. The *r_i_* of US frequency 60 kHz was greater than 0.5, and the difference in samples was small and the quality was good. The sample with an ultrasonic frequency of 20 kHz had the smallest *r_i_*, indicating that its quality was relatively poor. The difference in the quality of CP may be closely related to the ultrasonic frequency; high frequency US pretreatment reduced the drying time by 32.1% and improved the nutritional components of CP.

## 3. Materials and Methods

### 3.1. Materials

The CP used in the experiment was produced in Weiyuan, Dingxi, Gansu, China, and the harvest time was at the end of October. The two-year-old cultivated CP with an average diameter of (3 ± 0.5) cm, fresh, undamaged, and yellowish-brown, was selected as the test material. After purchase, it was stored at 5 ± 1 °C for no more than 2 weeks. The initial wet base moisture content of CP was 71.7 ± 0.5% (wb), which was determined at 105 ± 1 °C for 8 h.

### 3.2. Drying Pretreatment

The experiments were carried out in the KQ-300VDE three-frequency CNC ultrasonic cleaner (Kunshan Shumei Ultrasonic Instrument Co., Ltd., City, Kunshan, China). Add an appropriate amount of distilled water at 30 °C to the cleaning tank, and immerse the fresh CP into it. Based on the preliminary pretreatment experiments, ultrasonic frequencies (20, 40 and 60 kHz), ultrasonic power (150, 180 and 210 W) and ultrasonic treatment time (20, 30 and 40 min) were selected for the US pretreatment of CP. Before the test, the equipment needs to be adjusted to the preset parameters for preheating (temperature: 50 °C, irradiation height: 320 mm, vacuum: −20 KPa). To maintain a uniform thickness of each sample, fresh samples were sliced using a stainless steel slicer and then accurately measured with vernier callipers to an allowable thickness of 4 ± 0.05 mm. The total weight of the pre-treated slices was 120 ± 0.5 g, spread evenly in the drying tray and then placed in the WSH-60A vacuum far-infrared radiation multifunctional drying oven (Tianshui Shenghua Microwave Technology Co., Ltd., Tianshui, China) to complete the drying process with a weighing interval of 10 min. The drying process is completed when the moisture content is reduced to a safe moisture content of 10 ± 0.5%, corresponding to a dry basis moisture content of 0.10 ± 0.005 g/g. No ultrasonic pretreatment was carried out as a control group.

The procedure involved in this study is shown as Figure 7.

### 3.3. Calculation of Drying Characteristic Parameters

The data measured each time were used to draw the drying curve, including the moisture ratio and drying rate that change over time [42,43].

#### 3.3.1. Calculation of Dry Basis Moisture Content

The dry basis moisture content (*M_t_*) is calculated as follows:(1)$$M_{t}=\frac{W_{t}-G}{G},$$
where *W_t_* is the weight of CP slices at time *t*, g and *G* is the absolute mass of dry matter of CP slices, g.

#### 3.3.2. Calculation of the Moisture Ratio

Moisture ratio (*MR*) at different times is calculated as follows:(2)$$\;MR=\frac{M_{t}}{M_{0}},$$
where *M*_0_ is the initial dry basis moisture content of CP, g/g.

#### 3.3.3. Calculation of Drying Rate

The drying rate (*DR*) of CP in the drying process is calculated as follows:(3)$$\;DR=\frac{M_{t1}-M_{t2}}{t_{2}-t_{1}},$$
where *t*_1_, *t*_2_ are different drying times, min; *M_t_*_1_ and *M_t_*_2_ are the moisture content (in dry basis) of CP at time *t*_1_ and *t*_2_, respectively, g/g.

### 3.4. Effective Moisture Diffusion Coefficient

The moisture migration characteristics of the sample can be described using the effective moisture diffusivity, calculated by Fick’s second diffusion law mathematical model. The calculation formula is as follows [41]:(4)$$\;MR=\frac{M_{t}-M_{e}}{M_{0}-M_{e}}\approx \frac{M_{t}}{M_{0}}=\frac{8}{\pi^{2}}exp\left(-\pi^{2}\frac{D_{eff}t}{4L^{2}}\right),$$
where *D_eff_* is the effective moisture diffusivity, m^2^/min; *L* is half the thickness of the material, m; *t* is the time, min; and *M_e_* is the equilibrium moisture content, g/g.

For the calculation of *D_eff_* from the slope obtained by plotting ln*MR* versus drying time, the calculation formula is as follows:(5)$$\ln MR=\ln\frac{8}{\pi^{2}}-\frac{\pi^{2}D_{eff}}{4L^{2}}t,$$

### 3.5. Chemical Composition Analysis

#### 3.5.1. Preparation of CP Extract

Dried CP slices were crushed with a DLH-350A high-speed multifunctional grinder (Wuyi Qiteng Electric Appliance Co., Ltd., Jinhua, China) and then passed through a 60-mesh sieve. The sample powder (1.0 g) was placed in a cocked triangle bottle containing 25 mL 98% ethanol, extracted at room temperature and in darkness for 2 d, and oscillated continuously during the period; after centrifugation at 5000 r/min for 10 min, the extract was taken and stored at 4 °C for the determination of the total flavonoid, polysaccharide, total phenolic and antioxidant capacity of dried CP slices [44].

#### 3.5.2. Determination of Total Flavonoid Content

The total flavonoid content was determined with the sodium nitrite-aluminium nitrate-sodium hydroxide method [45]; a 2500 μL sample extract was placed in a test tube, and distilled water (2.0 mL) and 5% NaNO_2_ solution (0.3 mL) were added. After 0.3 mL 10% AlCl_3_ was added and mixed, then 2.0 mL 1 mol/L NaOH solution was added and mixed evenly. The absorbance (A) of the reaction solution was measured at 510 nm using a UV2600A UV-visible spectrophotometer (Unico Instrument Co., Ltd., Shanghai, China) while a solution without the sample was used as a blank control. Using Catechin (CE) as a standard, based on the linear relationship between the amount of the standard and the A value, the A-value of the sample solution was substituted into the equation to obtain the content. The total flavonoid content was determined and calculated as follows:(6)$$TFC=\frac{C_{a}V_{2}}{V_{1}M},$$
where *TFC* is the total flavonoid content, mg/g; *C_a_* is the amount of catechin, mg; *V*_1_ is the volume of sample extract for titration, mL; *V*_2_ is the total volume of the sample extract, mL; and *M* is the weight of CP dry matter, g.

#### 3.5.3. Determination of Polysaccharide Content

The content of total soluble sugar was determined using the phenol-sulfuric acid method [46]; 80 μL of the sample extract was added to 1.0 mL of phenol solution and 3.0 mL of concentrated H_2_SO_4_, it was reacted for 30 min, and the A at the wave length of 485 nm was measured. A standard curve for the polysaccharide content was obtained using Glucose as the standard. Based on the A values, the polysaccharide content was found on the standard curve and calculated as follows:(7)$$PC=\frac{C_{b}V_{2}}{V_{1}M},$$
where *PC* is polysaccharide content, mg/g; *C_b_* is the amount of glucose, mg.

#### 3.5.4. Determination of the Total Phenolic Content

The total phenolic compound content was determined using the Folin–Ciocalteau method [47]; 500 μL of sample extract was taken, and then 2.0 mL of 10% Folin–Ciocalteau and 1.0 mL of 7.5% Na_2_CO_3_ solution were added, respectively, the reaction was carried out at 37 °C in the dark for 60 min, and the A of solutions at 760 nm was determined. A standard curve for the total phenolic content was obtained using gallic acid (GAE) as the standard. Based on the A values, the total phenolic content was found on the standard curve and calculated as follows:(8)$$TPC=\frac{C_{s}V_{2}}{V_{1}M},$$
where *TPC* is the total phenolic content, mg/g; *C_s_* is the amount of gallic acid, mg.

#### 3.5.5. DPPH Radical Scavenging Assay

The determination of the DPPH scavenging capacity was carried out with the 1,1-diphenyl-2-picrylhydrazy1 (DPPH) method [48]. The 1250 μL sample extract was added to 3.0 mL 10^−4^ mol/L DPPH methanol solution, and the *A* was measured at 515 nm after 30 min dark vibration at room temperature. With 70% methanol as blank control and 500 μmol/L 90% ascorbic acid methanol solution as reference control, the absorbance was *A*_0_, and the absorbance of reaction solution was recorded. The DPPH scavenging activity of the CP slice to inhibition rate was an indicator. The inhibition rate calculation formula is as follows:(9)$$IR=\frac{A_{0}-A}{A_{0}},$$
where *IR* is inhibition rate, %; *A* is the absorbance of the sample solution; and *A*_0_ is the absorbance of the reference solution.

#### 3.5.6. Analysis of High-Performance Liquid Chromatography

The lobetyolin, atractylenolide III and syringin contents were determined using high performance liquid chromatography (HPLC) [49]. The analytical column was a (250 mm × 4.6 mm, 5 μm) Inertsil ODS C18 (Aglient, Palo Alto, CA, USA). The injection volume was 10 μL, with a column temperature of 25 °C, a wavelength of 268 nm and a flow rate of 1.0 mL/min. The mobile phase comprised water containing 1% acetic acid solution (solvent D) and methanol containing acetonitrile (solvent B), and was run in a 16 min gradient elution program [50]: 0–4 min, 15–40% B; 4–8 min, 40–65% B; 8–10 min, 65–85% B; 10–12 min, 85–15% B; 12–16 min, 15% B. Then, 1.00 mg of each of lobetyolin, atractylenolide III and syringin standards were mixed with methanol (1 mL) in a 10 mL tube. The mixture solution (0.3 mL) was diluted with methanol 0.7 mL to prepare a reference solution with a mass concentration of 300 μg/mL. The above control solutions were mixed and diluted to 6 different concentrations and used for linearity studies. Then, 1.00 g of CP sample powder was weighed precisely and dissolved in 25 mL of methanol solution, then processed in an ultrasonic cleaner, centrifuged for 10 min, and the supernatant filtered through a membrane for HPLC analysis [51].

Lobetyolin, atractylenolide III and syringin were quantified from their peak areas. The peak areas and mass concentrations of the mixed controls were used as standard curves.

### 3.6. Determination of Colour

The colour of CP was measured with a CR-410 colorimeter (Konica Minolta, Tokyo, Japan) under natural light for fresh and dry samples of CP; each group was measured three times and the average value was taken. The total colour difference, Δ*E*, represented the difference between the colour of the tested sample and the colour of the fresh sample; the smaller the Δ*E*, the better the quality of the dried CP. Δ*E* is calculated as follows [52]:(10)$$△E=\sqrt{{\left(L^{*}-L_{0}\right)}^{2}+{\left(a^{*}-a_{0}\right)}^{2}+{\left(b^{*}-b_{0}\right)}^{2}},$$
where *L***, a** and *b** are the brightness, red–green and blue–yellow values of the dried codonopsis product; *L*_0_, *a*_0_ and *b*_0_ are the brightness, red–green and blue–yellow values of the fresh CP.

### 3.7. Microstructural Analysis

In order to investigate the effects of different ultrasonic frequency, ultrasonic power and pretreatment time on the morphological characteristics of CP slices, the samples were sprayed with gold, and the samples were observed using a Hitachi-S3400 N scanning electron microscope (Hitachi Scientific Instruments Co., Ltd., Tokyo, Japan) at 5.0 KV acceleration voltage; the magnification was set to ×500 to obtain high-resolution images [53].

### 3.8. Evaluation of the Quality of CP

The grey correlation degree method was used to analyse and evaluate the quality of dried CP slices, and the relative correlation degree (*r_i_*) was used as the evaluation measure to construct the quality evaluation model of CP after US pretreatment [54,55]. There were n samples, and each sample had m evaluation indexes, thus forming the evaluation unit sequence {X_ik_} (i = 1, 2, 3,…, n; k = 1, 2, 3,…, m; in this study, n = 7, m = 10). When grey correlation degree is used as the evaluation measure, the reference sequence should be selected; generally, the optimal reference sequence and the worst reference sequence should be determined. If the optimal reference sequence is {X_sk_} (i = 1,2,3…n), the worst reference sequence is {X_tk_} (i = 1,2,3…n), where the indexes of the optimal reference sequence are the maximum values of the corresponding indexes of n samples, and the indexes of the worst reference sequence are the minimum values of the corresponding indexes of n samples.

The greater the correlation degree *r_i(s)_* relative to the optimal reference sequence, the smaller the correlation degree *r_i(t)_* relative to the worst reference sequence, indicating that the evaluation unit sequence is more ideal. The relative correlation degree *r_i_* of the defined evaluated unit sequence {X_ik_} relative to the optimal reference sequence {X_sk_} and the worst reference sequence {X_tk_} is calculated as follows [56]:(11)$$r_{i}=\frac{r_{i(s)}}{r_{i(s)}+r_{i(t)}}\;(i=1,2,\ldots n),$$

The sequences of evaluation units are ranked according to the magnitude of their relative correlation, and the final result of the merit evaluation is obtained.

### 3.9. Statistical Analysis

The experimental data were conducted using Excel 2020 and SPSS 26 software, and the experimental data were linearly fitted and plotted with origin 2018 software. Each group of experiments was repeated three times, and the mean and standard deviation of the results were calculated. One-way ANOVA was performed using Duncan’s multiple comparison test.

## 4. Conclusions

US pretreatment had significant effects on the drying rate, colour, microstructure and drying quality of CP slices. When the drying temperature was 50 °C and the US pretreatment parameters were 180 W, 60 kHz, 30 min, the drying time was 32.1% less than that without pretreatment. Compared with no pretreatment, the *D_eff_* value of ultrasonic pretreatment was between around 2.399 and 2.775 × 10^−8^ m^2^/min, an increase of 43.4%, the average content of polysaccharide and syringin increased by 14.7% and 62.0%, and the total flavonoid content decreased by 10.0%. The dried sample after US pretreatment showed good antioxidant properties. US pretreatment can significantly change the colour and brightness of the product after drying. The US pretreatment caused the formation of microchannels on the surface of cell structure of CP slices, which reduced the resistance of moisture evaporation and shortened the drying time. Thus, US pretreatment can significantly accelerate the drying rate of CP, shorten the drying time and improve the drying quality. The evaluation model established by the grey correlation degree could reflect the quality of CP slices more comprehensively. The US pretreatment conditions for the optimal quality of CP slices were 180 W, 60 kHz and 30 min. These results indicated that US can be used as a pretreatment technology to improve the processing efficiency and product quality of CP.

## Figures and Tables

**Figure 1 molecules-28-05596-f001:**
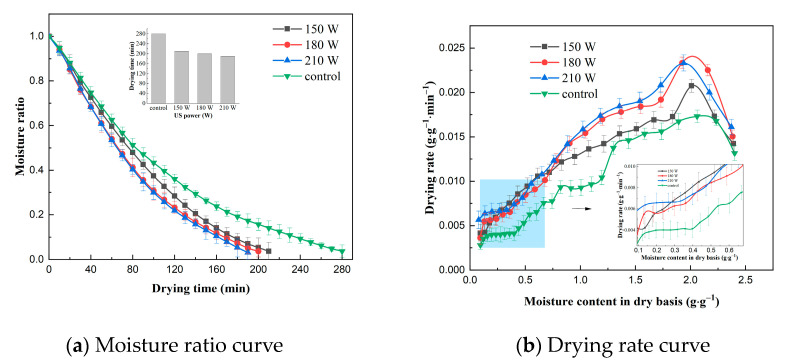
Effect of different ultrasonic frequencies on drying characteristics (control: without US).

**Figure 2 molecules-28-05596-f002:**
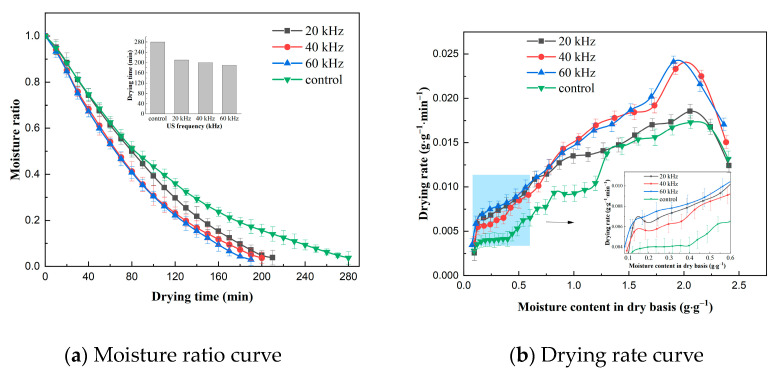
Effect of different ultrasonic power on drying characteristics (control: without US).

**Figure 3 molecules-28-05596-f003:**
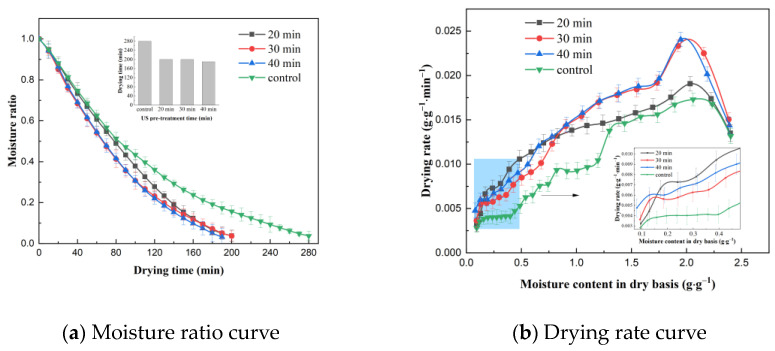
Effect of different pre-treatment times on drying characteristics (control: without US).

**Figure 4 molecules-28-05596-f004:**
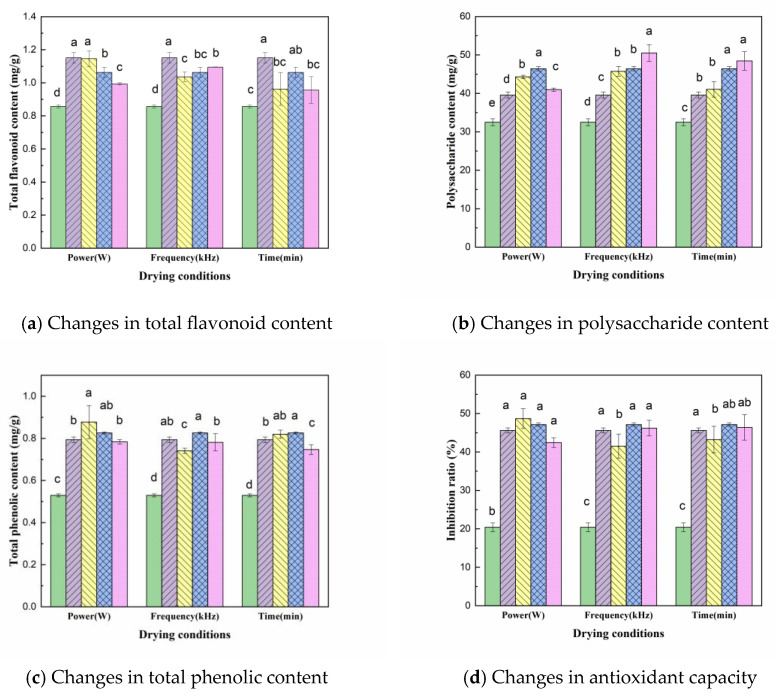
Changes in total flavonoid, polysaccharide, total phenolic and antioxidant capacity of CP under different drying conditions. Drying conditions: fresh (
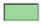
), control (
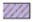
); Power: 150 W (
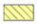
), 180 W (
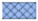
), 210 W (
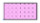
); Frequency: 20 kHz (
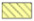
), 40 kHz (
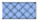
), 60 kHz (
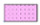
); Pre-treatment time: 20 min (
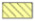
), 30 min (
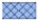
), 40 min (
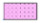
), Bars followed by the different letters (a–d) mean statistically different (*p* < 0.05).

**Figure 5 molecules-28-05596-f005:**
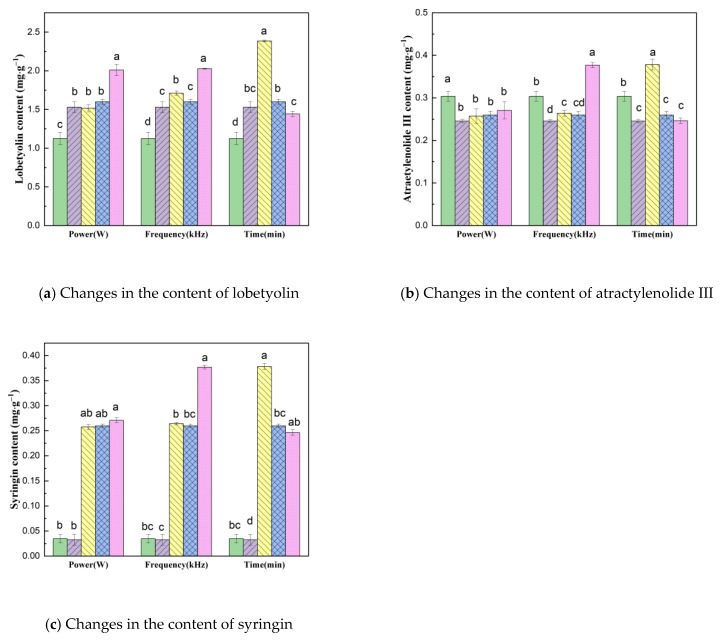
Changes in the content of CP lobetyolin, atractylenolide III and syringin under different pretreatment conditions. Drying conditions: fresh (
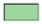
), control (
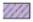
); Power: 150 W (
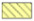
), 180 W (
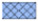
), 210 W (
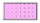
); Frequency: 20 kHz (
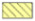
), 40 kHz (
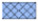
), 60 kHz (
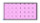
); Pre-treatment time: 20 min (
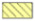
), 30 min (
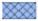
), 40 min (
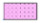
). Bars followed by different letters (a–d) mean statistically different (*p* < 0.05).

**Figure 6 molecules-28-05596-f006:**
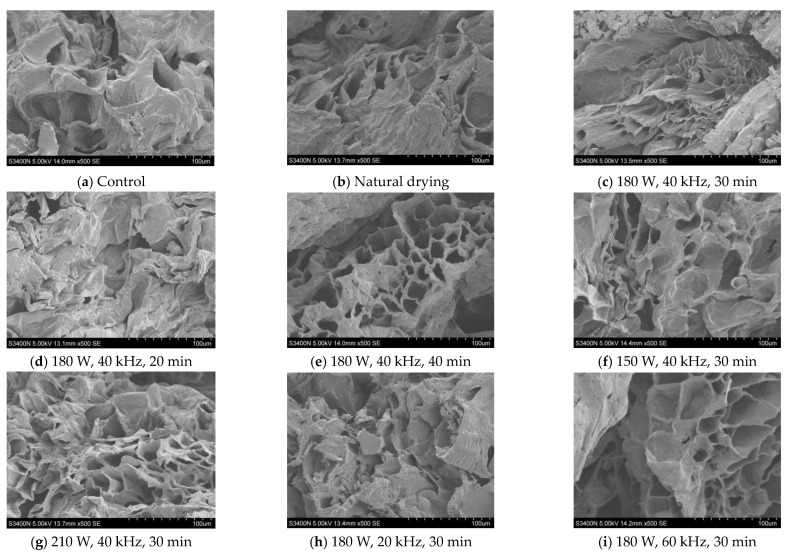
SEM images of dried CP at different pretreatment conditions.

**Figure 7 molecules-28-05596-f007:**
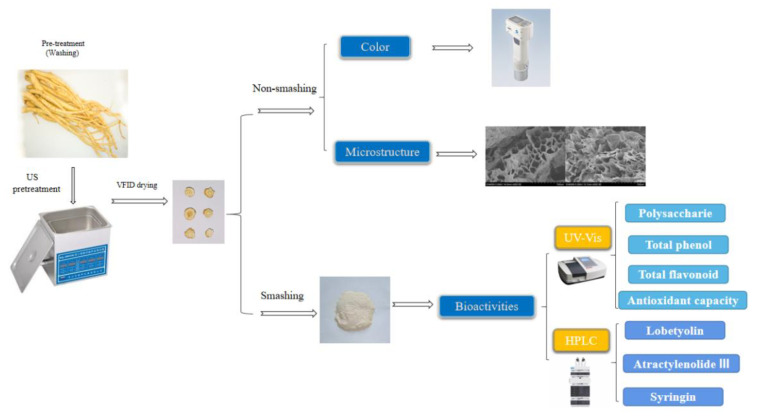
Process diagram of US pretreatment before VFID and quality detection of CP.

**Table 1 molecules-28-05596-t001:** Effective moisture diffusivity under different pretreatment conditions.

US Power/W	US Frequency/kHz	Pre-Treatment Time/min	Effective Moisture Diffusivity/m^2^·min^−1^·10^−8^
VFID	-	-	1.764
180	40	20	2.555
180	40	30	2.531
180	40	40	2.724
180	20	30	2.399
180	60	30	2.775
150	40	30	2.409
210	40	30	2.675

**Table 2 molecules-28-05596-t002:** Results of the examination of the linear relationships of the components (n = 6).

	Regression Equation	Linearity Range (mg/mL)	R^2^
Syringin	Y = 324.35009X + 23.54293	0.004~0.333	0.98271
Lobetyolin	Y = 60.83672X − 1.27109	0.004~0.333	0.99943
Atraetylenolide III	Y = 52.25826X − 3.00006	0.004~0.333	0.99951

**Table 3 molecules-28-05596-t003:** Effect of different pretreatment conditions on the colour of slices of CP.

Drying Method	*L**	*a**	*b**	Δ*E*
Fresh	68.98 ± 0.84 ^c^	−1.63 ± 0.04 ^f^	20.32 ± 0.92 ^a^	—
Control	60.58 ± 0.19 ^a^	0.48 ± 0.32 ^d^	14.93 ± 0.10 ^bc^	10.20 ± 0.04 ^a^
180 W−40 kHz−20 min	73.55 ± 0.68 ^b^	1.17 ± 0.35 ^bc^	15.92 ± 0.33 ^c^	6.93 ± 0.15 ^ab^
180 W–40 kHz–30 min	74.82 ± 0.23 ^a^	−0.68 ± 0.35 ^e^	17.43 ± 0.30 ^b^	6.58 ± 0.26 ^bc^
180 W–40 kHz–40 min	73.73 ± 0.71 ^b^	1.40 ± 0.46 ^bc^	20.70 ± 0.97 ^a^	5.64 ± 0.27 ^d^
180 W–20 kHz–30 min	73.77 ± 0.21 ^b^	2.41 ± 0.22 ^a^	19.66 ± 0.18 ^a^	6.30 ± 0.27 ^c^
180 W–60 kHz–30 min	74.75 ± 0.09 ^a^	0.98 ± 0.08 ^c^	17.93 ± 0.19 ^b^	6.76 ± 0.19 ^abc^
150 W–40 kHz–30 min	75.13 ± 0.04 ^a^	0.21 ± 0.43 ^d^	20.20 ± 0.87 ^a^	6.42 ± 0.07 ^bc^
210 W–40 kHz–30 min	73.83 ± 0.50 ^b^	1.55 ± 0.06 ^b^	17.91 ± 0.67 ^b^	6.28 ± 0.59 ^c^

*L**: brightness, *a**: red–green, *b^*^*: blue–yellow value. Note: Results are expressed as mean ± SE (n = 3). For each column, means followed by different letters are significantly different (*p* < 0.05).

**Table 4 molecules-28-05596-t004:** Sample dataset for CP.

Evaluation Units	Drying Time	*D_eff_*	Polysaccharide	Total Flavonoid	Total Phenolic	Antioxidant Capacity	Syringin	Lobetyolin	Atraetylenolide III	Colour
20 min	200	2.555	41.049	0.962	0.820	43.217	0.060	2.386	0.378	6.93
30 min	200	2.531	46.425	1.062	0.826	47.119	0.050	1.598	0.260	6.58
40 min	190	2.724	48.442	0.956	0.747	46.399	0.048	1.441	0.246	5.64
150 W	210	2.409	44.310	1.166	0.877	48.680	0.040	1.515	0.257	6.42
210 W	190	2.675	40.988	0.993	0.784	42.437	0.050	2.010	0.271	6.28
20 kHz	210	2.399	45.739	1.036	0.741	41.537	0.047	1.711	0.264	6.30
60 kHz	190	2.775	50.528	1.094	0.781	46.219	0.081	2.027	0.377	6.76

**Table 5 molecules-28-05596-t005:** Relative correlation and quality ranking of each sample.

Evaluation Units	Relative to the Optimal Reference Sequence *r_i(s)_*	Relative to the Worst Reference Sequence *r_i(t)_*	Relative Relevance *r_i_*	Quality Ranking
20 min	0.541	0.629	0.462	3
30 min	0.477	0.583	0.450	4
40 min	0.515	0.714	0.419	5
150 W	0.625	0.618	0.503	2
210 W	0.430	0.693	0.383	7
20 kHz	0.457	0.710	0.392	6
60 kHz	0.685	0.509	0.574	1

## Data Availability

The authors confirm that the data supporting the findings of this study are available within the article.

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
