# Peer review of "Evaluation of the Effect of Ultrasonic Pretreatment on the Drying Kinetics and Quality Characteristics of Codonopsis pilosula Slices Based on the Grey Correlation Method"

_molecules, 2023, doi:10.3390/molecules28145596_

Round 1
Reviewer 1 Report
This study was aimed to evaluate the effect of US pretreatment on the drying kinetics and quality characteristics of CP, including colour difference, microstructure, and functional composition. However, the writing of manuscript is not good, and some other questions and suggestions are as following.
1 please rewrite the last sentence in the abstract, and use the color or colure consistently in the whole manuscript.
2 in section 2.2.2, line 123, what’s the reason for centrifugation before absorbance measured at 512 nm?
3 section 3.3.4, please specify the antioxidant capacity with radical scavenging capacity.
4 in section 2.5.5, line 158-159, 90% ascorbic acid methanol solution as reference control, why? What’s the concentration of ascorbic acid?
5 please rewrite the section 2.5.6, and why you use the different extraction method for preparation of test sample solution from section 2.5 ?
6 for the bound phenolic fractions, did you identify the phenolic compounds?
7 section 2.7, a reference is suggested to be provided for scanning electron microscopy (SEM) analysis. https://doi.org/10.1016/j.lwt.2021.112740
9 section 4 should be conclusions.
Reviewer 2 Report
Detailed comments are included in the review file attached below.

Detailed comments are included in the review file attached below.
Round 2
Reviewer 1 Report
The manuscript has been improved according the suggestions.